# A multidimensional selective landscape drives adaptive divergence between and within closely related *Phlox* species

Benjamin E. Goulet-Scott [1,2], Matthew C. Farnitano [1,5], Andrea L. M. Brown [1,6], Charles O. Hale[1,7], Meghan Blumstein[3] & Robin Hopkins [1,4] ✉

Selection causes local adaptation across populations within species and simultaneously divergence between species. However, it is unclear if either the force of or the response to selection is similar across these scales. We show that natural selection drives divergence between closely related species in a pattern that is distinct from local adaptation within species. We use reciprocal transplant experiments across three species of *Phlox* wildflowers to characterize widespread adaptive divergence. Using provenance trials, we also find strong local adaptation between populations within a species. Comparing divergence and selection between these two scales of diversity we discover that one suite of traits predicts fitness differences between species and that an independent suite of traits predicts fitness variation within species. Selection drives divergence between species, contributing to speciation, while simultaneously favoring extensive diversity that is maintained across populations within a species. Our work demonstrates how the selection landscape is complex and multidimensional.

Ecological adaptation contributes to the origin and maintenance of biodiversity[1–4]. Evolution by natural selection drives local adaptation among populations within a species that occupy different biotic and abiotic environmental conditions[5–8]. Similarly, strong ecological selection can drive divergence between closely related species and cause reproductive isolation, leading to ecological speciation[1,9,10]. Although extensive research has investigated local adaptation within species and investigated ecological divergence between species, little is known about if and how these evolutionary processes are related. Are the axes of selection favoring adaptive divergence between species the same or different than the axes of selection favoring local adaptation within a species? Characterizing local adaptation both within and between closely related species can offer insight into how

ecological adaptation generates diversity from the micro- to macro-evolutionary scale.

Disparate populations within wide-ranging species often evolve to become adapted to the local ecological conditions[7]. Across the tree of life, there are striking examples of variation in morphology, physiology, and phenology within species across populations that span significant gradients of temperature, water availability, seasonality, and types of biotic interactions[11–16]. Although very common, local adaptation among populations within a species is not inevitable. The extent of divergence can depend on the relative strengths of selection and migration[17], the presence or absence of fitness trade-offs in different habitats[6], and the genetic correlation between traits[18–20]. Nevertheless, meta-analyses conclude that adaptive divergence

[1]Department of Organismic and Evolutionary Biology, Harvard University, Cambridge, MA 02138, USA. [2]Harvard Forest, Harvard University, Petersham, MA 01366, USA. [3]Civil and Environmental Engineering, Massachusetts Institute of Technology, Cambridge, MA 02139, USA. [4]Arnold Arboretum of Harvard University, Boston, MA 02131, USA. [5]Present address: Department of Genetics, University of Georgia, Athens, GA 30602, USA. [6]Present address: Department of Environmental Science, Policy, and Management, University of California Berkeley, Berkeley, CA 94720, USA. [7]Present address: Institute for Genomic Diversity, Cornell University, Ithaca, NY 14853, USA. ✉e-mail: rhopkins@fas.harvard.edu

between populations of a species is widespread in nature and maintained despite gene flow between populations[6,21].

As taxa become reproductively isolated, they tend to evolve suites of diverged traits that lead to higher fitness (survival or reproduction) in local or native habitats compared to habitats of closely related taxa. Adaptive divergence often leads to a similar pattern of reciprocal local vs. foreign advantage between closely related species as the pattern that we see between locally adapted populations within a single species. Even when closely related species are in broad sympatry with extensive geographic overlap, we may expect a pattern of adaptive divergence. Interspecific competition for resources can select for ecological divergence and niche partitioning driving either species-wide patterns of differentiation or leading to patterns of character displacement in sympatry[2,22]. Therefore, we expect many of the traits that differentiate species from each other to be the suites of traits that lead to differential fitness and thus ecological reproductive isolation between the species. In this way, adaptive divergence not only causes phenotypic differentiation between taxa but also contributes to the cessation of gene flow between taxa, leading to speciation. For this reason, ecological adaptation is considered important during the speciation process[23]. In fact, environmental divergence and ecological divergence are often added to, and portrayed as parallel to, the speciation continuum from no reproductive isolation to complete reproductive isolation[24,25].

The idea of a continuum of adaptive ecological divergence -- from producing and maintaining diversity within species to causing reproductive isolation between species -- invites us to consider how patterns of adaptation within and between species may or may not be related[2,23,26]. Under one notion of this continuum, the adaptive divergence that we see between species is an extreme case of the local adaptation we see within species across populations and therefore could be due to the similar axis of selection and involve similar types of trait divergence (Fig. 1, top & bottom right). Alternatively, the types of

selection driving divergence between species could be distinct from the selection pressures favoring local adaptation within a species (Fig. 1, top & bottom left). Under this latter scenario, the trait divergence that differentiates species is different from the variation that we see within a species. Importantly, ecological divergence and speciation unfold over evolutionary time, and the snapshot of divergences we see now between populations and species does not directly tell us about how the process of speciation did or will proceed in this system[25]. Nonetheless, comparing patterns of phenotypic divergence and axes of selection across phylogenetic scales can help us understand how phenotypic diversity is generated and maintained under different scales of geographic range and genetic exchange.

*Phlox pilosa* subsp. *pilosa* (hereafter "pilosa"), *P. amoena* subsp. *amoena* (amoena), and *P. pilosa* subsp. *deamii* (deamii) are three closely related perennial wildflower taxa inhabiting the eastern U.S. that provide a promising system in which to evaluate patterns of ecological differentiation, both within and between species[27]. The three species have strikingly similar floral traits although distinctive vegetative characteristics[27,28]. The ranges of these three *Phlox* taxa overlap in western Kentucky, Tennessee, and Indiana, but they rarely co-occur in the same locality, suggesting differences in habitat preference[27-29]. Here, we use a combination of reciprocal transplant[30,31] and provenance trial[32-34] approaches to evaluate the presence and strength of local adaptation between and within species. Specifically, we: (1) model and compare the ecological niches of the *Phlox* species; (2) determine whether there is an adaptive divergence between the three species; (3) infer if there is local adaptation within *Phlox* species; and (4) evaluate patterns of phenotypic diversity across all three species and compare axes of selection driving divergence between and within species. Collectively, this study provides unique insights into how selection operates to drive diversity across scales of micro- and macro-evolution.

## Results
### Ecological niche modeling
We built ecological niche models for the two widespread *Phlox* species, amoena, and pilosa, using available occurrence data and biologically relevant environmental variables (Fig. 2A, Fig. S1, Supplementary Data Tables 1 and 2). The predicted extents of suitable habitat conform well to the described geographic ranges of these species with a broad range of sympatry from Georgia to Kentucky[28,29].

Deamii is a relatively rare endemic with only 5 documented occurrences. We included this closely related species in our study to better understand broad patterns of adaptive divergence but were unable to build an ecological niche model for deamii or test local adaptation within species due to the low number of known occurrence points. It is hypothesized that deamii populations experience a narrow range of environmental conditions and are broadly sympatric with both amoena and pilosa[28].

From a principal component analysis (PCA) of the environmental variables used to build our niche models, we find that pilosa inhabits a greater breadth of ecological variation than does amoena (Fig. 2B). While both species occupy a similar amount of variation on PC2, amoena occupies a subset of the variation covered by pilosa on PC1. We find that the median conditions occupied by amoena and pilosa are significantly different on PC1 but not on PC2 (Fig. S1C). Of note, the common garden sites chosen to represent amoena and pilosa habitats in our reciprocal transplant experiment described below differ along PC1 as well (colored diamonds in Fig. 2B). The reciprocal transplant experiment includes individuals sampled from populations that reasonably encompass the environmental variation experienced by these species (black edged circles Fig. 2B, Supplementary Data Table 5).

### Adaptive divergence between taxa
We find strong evidence of adaptive divergence between *Phlox* species from our reciprocal transplant experiments. Multiple individuals

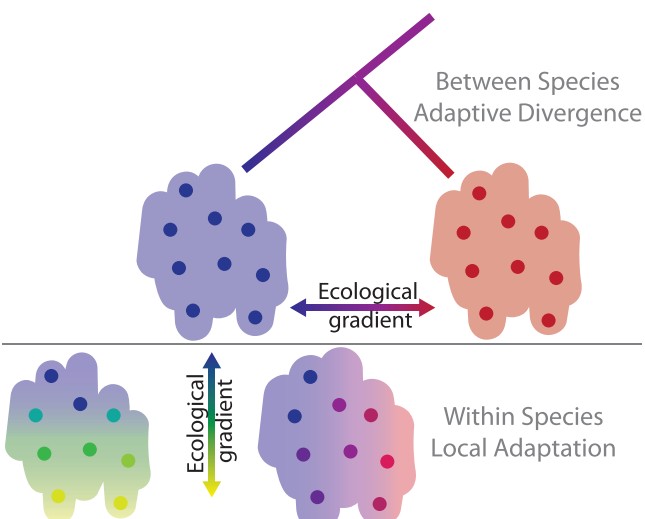

**Fig. 1 | Conceptual schematic representing divergence across scales of biological diversity in response to selection along axes of ecological variation.** Top panel represents the adaptive divergence between populations of two species shown as blue and red dots on different ecological habitats denoted by red and blue backgrounds. Bottom panel represents alternative scenarios of within-species local adaptation. Each colored point is a population adapted to the gradient of ecological conditions in the habitat represented by color across the background. In the scenario shown at the right (blue to red), the ecological gradient driving within-species local adaptation is parallel to the ecological gradient driving between species adaptive divergence. In the left scenario (blue to yellow) the gradient of within-species adaptation is orthogonal to the gradient driving divergence between species (blue to yellow).

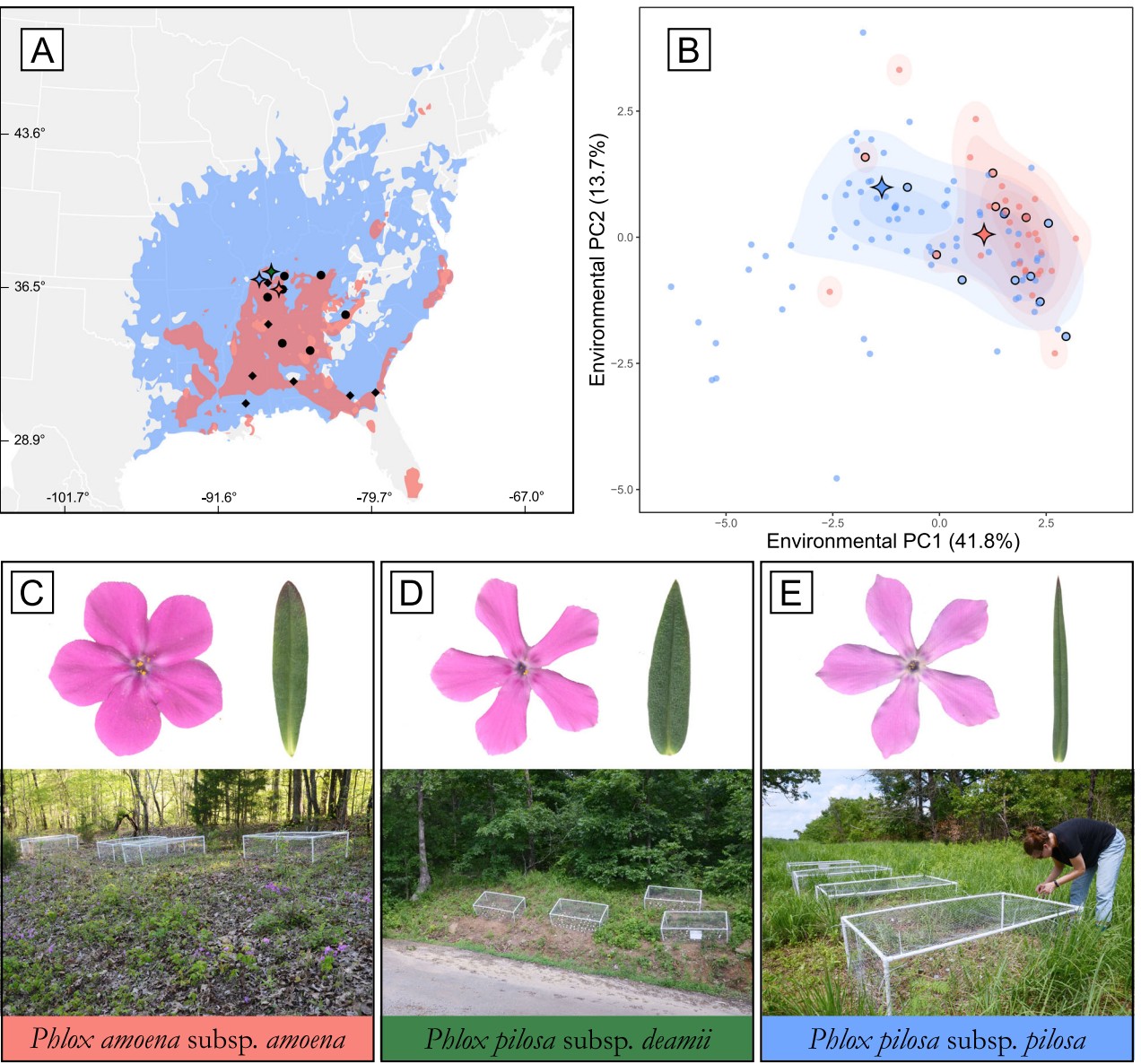

**Fig. 2 | Geographic and environmental variation of broadly sympatric *Phlox* species. A** Ecological niche modeling predicts the geographic distributions of *P. pilosa pilosa* (pilosa; blue) and *P. amoena amoena* (amoena; red) across eastern North American (longitude and latitude indicated) with sampling locations shown as black diamonds (pilosa) and black circles (amoena). Locations of the common gardens are indicated by colored diamonds (amoena in red, pilosa in blue, deamii in green). **B** Environmental variation of pilosa and amoena summarized with a

principal component analysis. Blue and red points indicate conditions of known populations of pilosa and amoena respectively. Black outlined points are populations sampled for transplant experiments and diamonds are the common garden sites. Representative flowers and leaves (not to scale) and pictures of local common garden site, of amoena (**C**), *P. pilosa deamii* (deamii) (**D**), and pilosa (**E**). Pictures taken by author B.E. Goulet-Scott.

sampled from source populations throughout the ranges of these three perennial *Phlox* species (black diamonds and circles Fig. 2A) were clonally replicated into common gardens in the native sympatric range of these species. Our experiment included three garden sites each adjacent to a wild population of one of the focal taxa (Fig. 2C–E, Supplementary Data Table 6). We quantified five fitness-related traits: herbivory, fruit number, flower number, biomass, and survival, and find the relative success of a species depends on the garden in which they are grown, as indicated by statistical support for a taxon-by-garden interaction (Fig. 3; Table 1). Adaptive divergence is evidenced by either the local species having higher fitness than the foreign species in the local species' garden, or by a focal species having the highest fitness in its home garden compared to all other away gardens.

All significant local vs. foreign comparisons match the prediction of adaptive divergence between taxa with the local taxon out-performing the foreign taxa (Table 1, Fig. 3). In the amoena habitat, amoena had nearly twice the survival as compared to deamii and 1.5 times the survival of pilosa. Amoena also experienced a third to a half as much major herbivory as pilosa and deamii, and produced more fruits than pilosa plants. In the deamii habitat, deamii survived nearly three times more than pilosa. In the pilosa habitat, pilosa plants produced three times as many fruits and survived twice as much as amoena and deamii plants. Effect size estimates for each contrast are illustrated in Fig. 3F.

Home vs. away comparisons (comparing across habitats for each taxon) showed some significant differences in the direction predicted by adaptive divergence (Table 1). Pilosa had the highest fitness in the

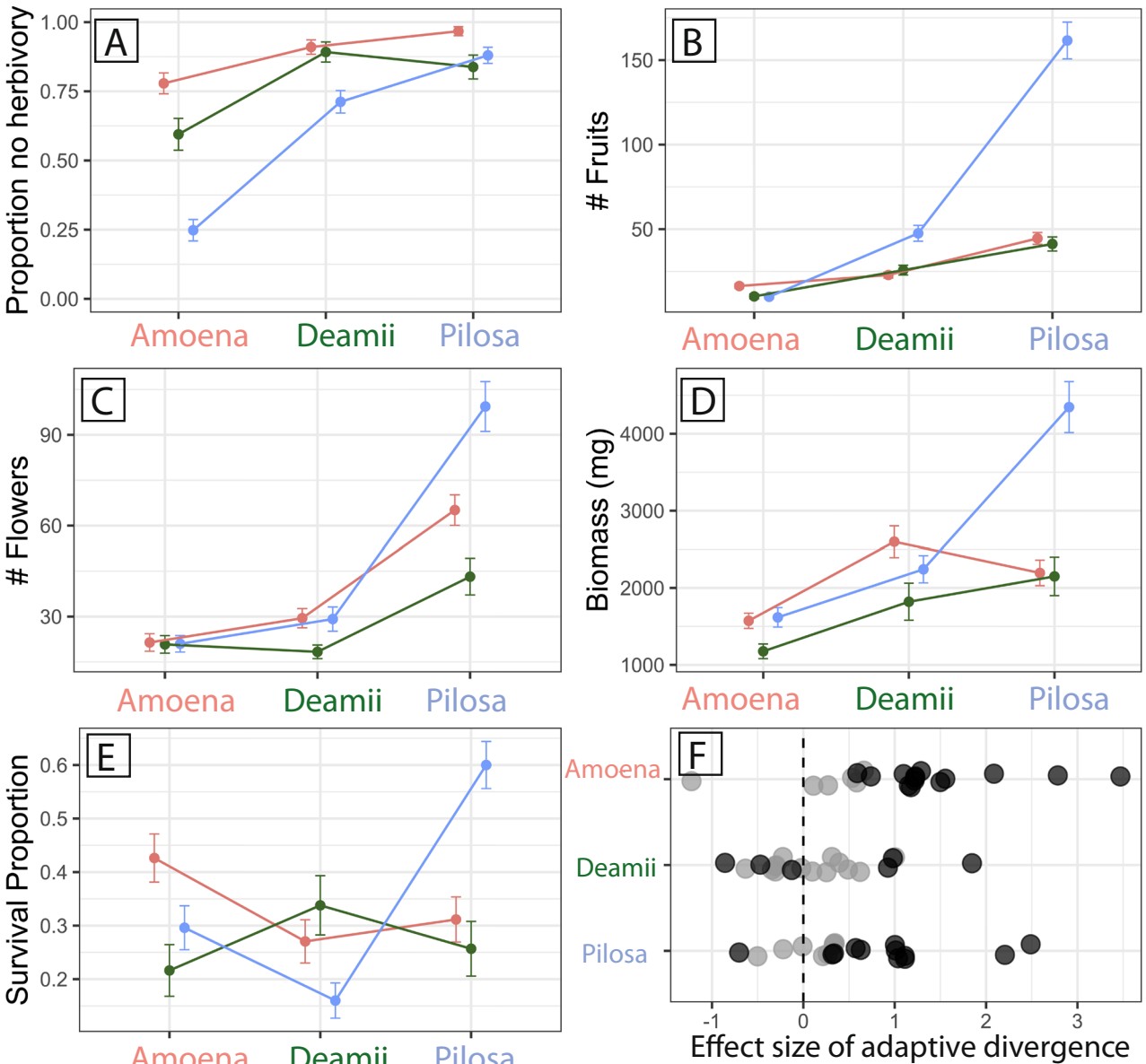

**Fig. 3 | Performance of each taxon across three garden environments.** Fitness traits include **A** proportion of plants without herbivore damage, **B** total number of fruits, **C** total number of flowers, **D** aboveground biomass, and **E** proportion survived to the end of the experiment. Values plotted are taxon means ± standard error in each garden ($n = 321$ individuals per garden). The ANOVA evaluation of a mixed model analysis for each trait revealed a significant taxon-by-garden interaction for all traits. **F** Summary of the effect size of post-hoc contrasts evaluating local adaptation and home-garden advantage for each species. Positive values indicate local species performed superior while negative values indicate local species performed worse. Black points indicate Tukey Test contrasts are significant at $p < 0.05$. See Table 1 for full model results.

home garden compared to in the other gardens on all five fitness traits. Deamii had less herbivory and set more fruits in the home garden compared to the amoena garden. We also found some patterns of success that did not indicate the highest success at home sites. For instance, deamii and amoena had fewer flowers and fruits in their home gardens compared to either of the other gardens.

## Local adaptation within species

We find strong evidence of local adaptation across populations within pilosa. We used statistical models to estimate the contribution of the source population to variation in the five fitness-related traits for amoena and pilosa. Local adaptation was evidenced by a negative relationship between the estimated population effect on fitness and the distance of the population from the common garden. This relationship was tested for geographic distance, genetic distance

(as measured by $F_{ST}$ using data from Goulet-Scott et al.[27]), and environmental distance (as measured in climate PC space) between populations (Supplementary Data Tables 3 and 4).

For the pilosa species within the pilosa garden, local adaptation was evidenced by a negative relationship between the estimated population effect on flower and fruit number fitness traits and geographic distance, environmental distance, and genetic distance. Final biomass in pilosa also shows a strong negative correlation with geographic distance (Fig. 4; Supplementary Data Table 4). Specifically, we estimate that biomass decreases by a milligram per kilometer distance between the source and the common garden (Supplementary Data Table 4). In the amoena habitat, pilosa populations also show strong negative correlations between biomass and geographic distance and are similarly predicted to lose a milligram of biomass per kilometer distance from the garden. Pilosa produces fewer flowers in the amoena

**Table 1 | Model details with contrast estimates for five fitness-related traits measured in a three-garden reciprocal transplant experiment**

| | taxon*garden | amoena | | deamii | | pilosa | |
|---|---|---|---|---|---|---|---|
| | | local vs. foreign | home vs. away | local vs. foreign | home vs. away | local vs. foreign | home vs. away |
| **herbivory** | X² = 17.87 | vs. deamii = −1.03, p = 0.037 | vs. Dea = 1.11, p = 0.005 | vs. amoena = −0.30, p = 0.625 | vs. Amo = 1.85, p < 0.001 | vs. amoena = −1.22, p = 0.06 | vs. Amo = 3.47, p < 0.001 |
| | p = 0.001 | vs. pilosa = −2.49, p < 0.001 | vs. Pil = 2.21, p < 0.001 | vs. pilosa = −1.00, p = 0.082 | vs. Pil = −0.49, p = 0.328 | vs. deamii = 0.66, p = 0.25 | vs. Dea = 1.17, p < 0.001 |
| **flower #** | X² = 685.89 | vs. deamii = 0.22, p = 0.703 | vs. Dea = 0.32, p < 0.001 | vs. amoena = −0.22, p = 0.698 | vs. Amo = −0.13, p < 0.001 | vs. amoena = 0.11, p = 0.789 | vs. Amo = 1.55, p < 0.001 |
| | p < 0.001 | vs. pilosa = −0.33, p = 0.422 | vs. Pil = 1.11, p < 0.001 | vs. pilosa = −0.10, p = 0.864 | vs. Pil = 0.86, p < 0.001 | vs. deamii = 0.27, p = 0.634 | vs. Dea = 1.23, p < 0.001 |
| **fruit #** | X² = 2583.5 | vs. deamii = −0.33, p = 0.307 | vs. Dea = 0.34, p < 0.001 | vs. amoena = 0.25, p = 0.439 | vs. Amo = 0.92, p < 0.001 | vs. amoena = 1.15, p < 0.001 | vs. Amo = 2.78, p < 0.001 |
| | p < 0.001 | vs. pilosa = −0.63, p = 0.007 | vs. Pil = 1.00, p < 0.001 | vs. pilosa = 0.34, p = 0.288 | vs. Pil = 0.47, p < 0.001 | vs. deamii = 1.10, p < 0.001 | vs. Dea = 1.22, p < 0.001 |
| **biomass** | X² = 22.03 | vs. deamii = −0.34, p = 0.384 | vs. Dea = 0.27, p = 0.068 | vs. amoena = −0.63, p = 0.110 | vs. Amo = −0.02, p = 0.91 | vs. amoena = 0.53, p = 0.07 | vs. Amo = 0.74, p < 0.001 |
| | p < 0.001 | vs. pilosa = −0.21, p = 0.466 | vs. Pil = −0.01 Pil, p = 0.955 | vs. pilosa = 0.30, p = 0.4425 | vs. Pil = 0.31, p = 0.103 | vs. deamii = 0.58, p = 0.135 | vs. Dea = 0.59, p = 0.001 |
| **survival** | X² = 43.05 | vs. deamii = −1.01, p = 0.004 | vs. Dea = −0.70, p = 0.011 | vs. amoena = 0.31, p = 0.355 | vs. Amo = 0.62, p = 0.099 | vs. amoena = 1.22, p < 0.001 | vs. Amo = 1.29, p < 0.001 |
| | p < 0.001 | vs. pilosa = −0.57, p = 0.041 | vs. Pil = −0.50, p = 0.063 | vs. pilosa = −0.98, p = 0.007 | vs. Pil = −0.39, p = 0.279 | vs. deamii = 1.50, p < 0.001 | vs. Dea = 2.09, p < 0.001 |

The X² and p-values reported for taxon*garden interactions were determined by ANOVA on generalized linear mixed models as described in the "Methods" section. The contrast effect size estimates reported for local vs. foreign and home vs. away comparisons were determined using Tukey's Test. Dea indicates the deamii home garden site, Pil indicates the pilosa home garden site, and Amo indicates the amoena home garden site.

garden as all three distances increase from the garden. (Supplementary Data Table 4). In the amoena garden, pilosa populations show a strong positive correlation between the proportion of plants without herbivore damage and both genetic and geographic distance; this represents the only signal in our data that does not support local adaptation. Within the deamii garden, pilosa populations show a negative correlation between the number of flowers produced and geographic distance while amoena populations show negative correlations between survival and both genetic and environmental distance. There was insufficient variation in herbivory among amoena populations and survival among pilosa populations to model population effects suggesting no evidence for local adaptation in these two traits.

**Selection between and within species**

Adaptive differentiation between *Phlox* species and local adaptation within species occur along different axes of variation in leaf morphology and physiology, likely driven by different axes of selection. For each individual genotype used in the experiment, we measured or calculated six phenotypic traits including leaf length, leaf width, leaf length/width ratio, leaf area, leaf chlorophyll content, and specific leaf area (SLA). Due to collinearity between traits, we summarized phenotypic variation using a principal components analysis of the trait measurements (Fig. 5). We used a series of regression models to investigate how leaf trait variation (as described by PC1 and PC2) explained variation in normalized fitness (fruit set, flower set, and biomass normalized to the average of each trait) between and within species in the pilosa habitat common garden (Fig. 5, Supplementary Data Tables 8 and 9). We then transform our findings about PC variation and fitness back onto our leaf traits.

The principal components analysis summarizing the phenotypic variation across species sharply divides pilosa from amoena and deamii individuals along PC1 (Fig. 5A). Taxon identity explains 72% of the variation along this first principal component ($F(2318) = 417.34$, $p < 0.001$). PC1 explains 45% of trait variation and describes leaf shape (Supplementary Data Table 7). Long narrow leaves and low chlorophyll content are at one end of the PC axis (pilosa-like), and short wide leaves with high chlorophyll content are at the other (amoena-like). All species show extensive and overlapping variation along PC2, which explains 27.8% of the phenotypic variation and corresponds to variation in the size of the leaf (area and SLA) (Supplementary Data Table 7). Taxon identity explains none of the variations along PC2 ($F(2318) = 0.8$, $p = 0.451$).

We considered fitness variation due to PC1 and PC2 using two sets of models. First, we modeled variation in fitness traits as explained by each trait PC while controlling for taxon and the interaction between taxon and trait PC. For PC1, we found that taxon identity predicted fitness-related traits consistent with our tests of adaptive divergence previously discussed. Due to the collinearity between taxon identity and value at PC1, this trait PC is not significant in our model when controlling for taxon (Supplementary Data Table 8). Pilosa individuals have both higher values along PC1 and high fitness in the pilosa garden. For PC2 the strength and direction of selection varied across species as indicated by the significant interaction term in our model (Supplementary Data Table 8).

With our second set of models, we evaluated how each leaf trait PC predicts fitness traits across all the species and within each of the species (Supplementary Data Table 9). PC1 does not predict fitness variation within any of the three species; it is only when individuals from all three species are included in the model together that we see a significant relationship between PC1 and fitness-related traits (Fig. 5B, D, F, Supplementary Data Table 9). In contrast, we find that within pilosa and amoena PC2 strongly predicts fitness traits and that this variation explains the significant relationship between PC2 and fitness in the combined dataset (Fig. 5C, E, G, Supplementary Data Table 9). Together our models indicate that leaf shape (PC1) differs

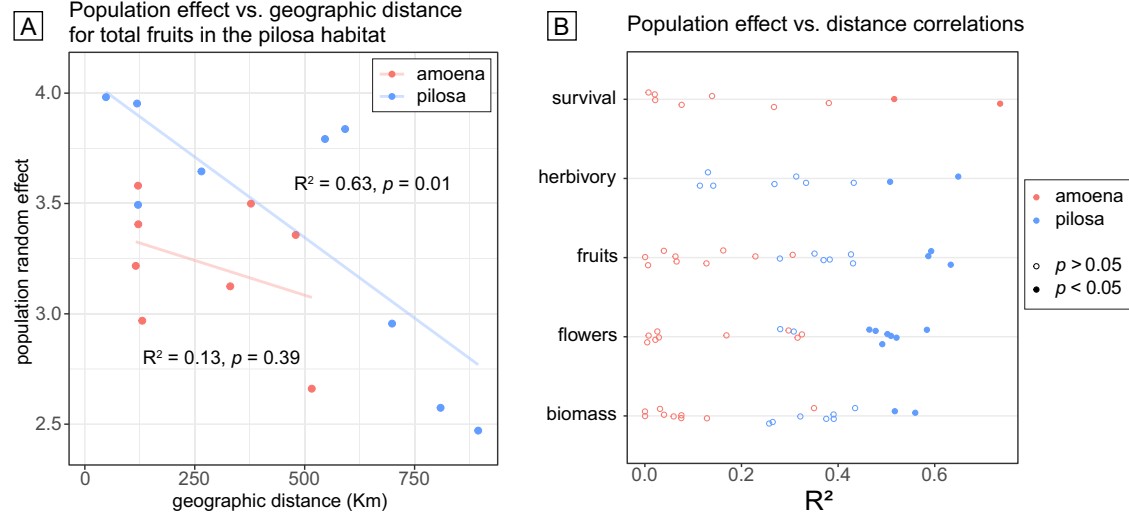

**Fig. 4 | Distance from the common garden predicts *P. pilosa pilosa* (pilosa) success indicating local adaptation.** As an example, **A** the relationship between population effect on total fruit set success and geographic distance for *P. amoena amoena* (amoena; red) and *P. pilosa pilosa* (pilosa; blue) populations with $R^2$ values from regression models indicated. **B** Distribution of $R^2$ values for regression models of population effect vs. distance measures among populations of amoena (red) and pilosa (blue) grown in all three experimental gardens. Solid points indicate significant evidence of local adaptation, where $p < 0.05$ in an F-statistic hypothesis testing of the model. Full model details are in Supplementary Data Table 4.

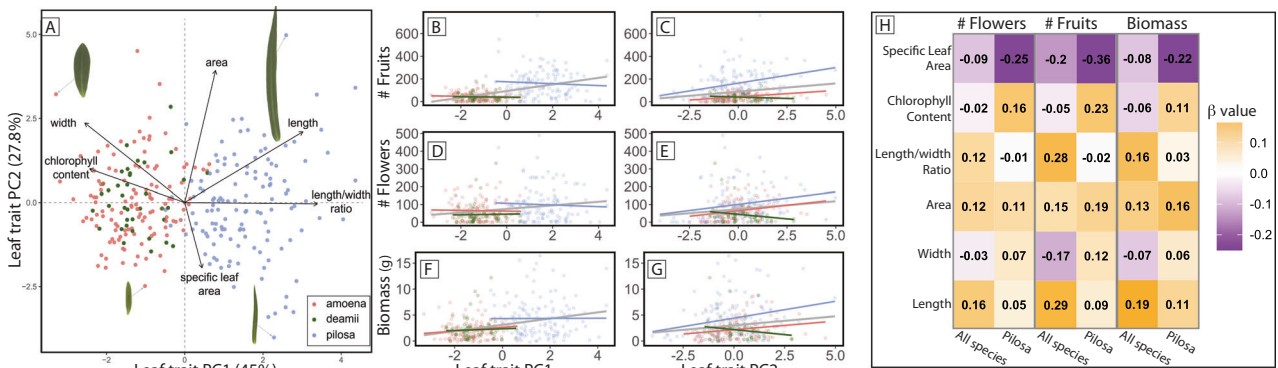

**Fig. 5 | Trait variation predicts fitness variation between and within species.** Principal components analysis describing phenotypic variation across three *Phlox* species (**A**). Points indicate values from individuals grown in the common garden experiment and black arrows indicate loadings of specific traits on the axes of variation. Images of four leaves (to scale) are connected to their points with gray arrows to demonstrate variation in shape along PC1 and area along PC2. **B–G** Relationship between fitness traits and leaf trait variation along PC1 and PC2. Colored points indicate the fitness values of individuals grown in the pilosa habitat garden. Gray lines indicate the linear model found significant relationships across all species and colored lines show relationships within each species (pilosa in blue, amoena in red, and deamii in green). For PC1, linear models find no relationships within species are significant while all within pilosa and amoena relationships are significant between PC2 and fitness traits. Full results in Supplementary Data Table 8. **H** Heatmap of reconstituted selection gradients (β values) for three fitness measures on each leaf trait with darker colors indicated higher values, negative values in purple, and positive values in orange. Results are shown for data from all species and just for pilosa for each fitness measure. β values are included in each well.

significantly between *Phlox* species and it is therefore these differences that correspond to fitness differences *between* species in the common garden. In contrast, leaf size (PC2) varies within species and significantly predicts *within*-species variation in fitness in the common garden (Supplementary Data Table 9).

Although the patterns in our models indicate that different traits underly fitness differences between species compared to fitness differences between populations, measuring selection on PC scores can be difficult to interpret. To overcome the problem of biologically interpreting PC scores, we transform selection gradients for the PC scores back into the original traits[35]. This method multiplies the matrix of eigenvectors from the leaf trait PCA (Supplementary Data Table 7) by the vector of regression coefficients of normalized fitness on the first three PC scores (Supplementary Data Table 11) to generate a vector of reconstructed selection gradients (Fig. 5H). We performed this analysis using data from all species grown in the pilosa garden and for only pilosa individuals in the garden. The results reveal that fitness differences between species are due to selection acting on leaf length ($\beta_{flower\#} = 0.22$, $\beta_{fruit\#} = 0.46$, $\beta_{biomass} = 0.2$), and leaf length/width ratio (leaf shape) ($\beta_{flower\#} = 0.16$, $\beta_{fruit\#} = 0.44$, $\beta_{biomass} = 0.17$), whereas within pilosa the strongest selection is acting on specific leaf area ($\beta_{flower\#} = -0.35$, $\beta_{fruit\#} = -0.56$, $\beta_{biomass} = -0.23$), leaf area ($\beta_{flower\#} = 0.15$, $\beta_{fruit\#} = 0.29$, $\beta_{biomass} = 0.16$), and chlorophyll content ($\beta_{flower\#} = 0.23$, $\beta_{fruit\#} = 0.37$, $\beta_{biomass} = 0.12$). The patterns of selection gradients across three proxies for fitness (fruit number, flower number, and biomass) all indicate that the strength of selection and even the direction of selection is different within versus between species.

## Discussion

Natural selection is widely acknowledged as the most important force underlying the evolution of biological diversity, yet we still have much to learn about how this process acts across micro- and macroevolutionary scales. At one end of this scale, populations within a species can locally adapt in response to variation in selection across space, generating diversity within a species; while at the other end, response to selection can drive adaptive divergence between taxa and even cause significant reproductive isolation, thus contributing to speciation. Many studies have characterized the response to selection at one scale or the other, yet there are few studies that integrate across scales to compare how selection simultaneously drives divergence within and between species.

Here we have characterized adaptive divergence between closely related species and local adaptation within one of these species. Furthermore, we show that selection-driving divergence between species is distinct in strength and direction from selection-driving divergence within species. Our results suggest a broadly applicable explanation of how a species can both maintain extensive adaptive phenotypic variation across broad ecological habitats while simultaneously maintaining distinct adaptive divergence from recently diverged taxa. Selection acts along many axes and the axis correlated with reproductive isolation and species interaction may be entirely different from the axis allowing populations to locally adapt across a species' range.

Our results suggest that natural selection drove adaptive divergence between the three co-occurring species of *Phlox* – pilosa, amoena, and deamii. The widespread species – pilosa and amoena – show broadly sympatric ranges with statistically significant yet minimal niche divergence. Specifically, amoena inhabits a distinct subset of the broader environmental tolerance of pilosa, likely reflecting the more northern range limit of pilosa compared to amoena. The patterns we observed in these closely related *Phlox* species are likely similar to many wide-ranging species. Ecological niche modeling that focuses on environmental conditions such as temperature and precipitation can characterize overlapping niches for species that are never found growing together but have broadly overlapping ranges. Due to this significant overlap in both geographic and environmental space, our niche modeling may suggest minimal adaptive divergence between species, and yet our experimental gardens reveal extensive fitness differences.

Across the five proxies for fitness we measured, we found that the local species generally does better in its local habitat garden as would be predicted by adaptive divergence between species. Because we measured five traits in three gardens across three species, we performed abundant statistical tests to identify patterns of differential success, which likely led to some false positives. We focus not on the results from any specific test but instead on the robust pattern that, for each species, we found evidence for natural selection favoring the local species. The specific patterns of adaptive divergence are different for each species, which is consistent with other studies that find that different lineages locally adapt in different ways[16]. For example, in the amoena garden, there was extensive large-mammal herbivory with nearly 50% of the plants showing signs of severe damage, but amoena plants suffered the least damage and the highest survival. The pilosa garden had the greatest sun exposure and the pilosa plants seem to exploit this light to have the highest survival and set the most fruits. Although our conclusions of adaptive differentiation are strongly supported, this work inspires future investigations to untangle the specific selective agents and traits underlying this pattern.

The support for adaptive differentiation between species may be particularly surprising given that the three common gardens were geographically close (within 120 km of each other) but the individuals in the garden were sourced from across the country, spanning 900 km. The patterns of adaptation were robust to the extensive geographic sampling and the breadth of source environmental conditions.

This suggests that the traits that adaptively differentiate the species are shared across populations within their ranges and could therefore contribute to ecological reproductive isolation between species.

As is often observed for widespread species, one of our *Phlox* species also shows strong patterns of local adaptation among populations. Two of our *Phlox* species span extensive environmental gradients with large (and overlapping) geographic ranges. This presents the opportunity for selection to favor different trait values between, for example, the warm and dry habitats in northern Florida and the cooler and wetter populations in western Kentucky. If local adaptation within species is driven by these ecological gradients across their ranges, then we predict that as distance increases between the population source and an experimental garden, fitness will decrease. This is precisely the pattern we documented across pilosa populations. Individuals sourced from populations near the pilosa experimental garden grew bigger, had more flowers, and set more fruits than individuals from populations farther away from the experimental garden. This signal was robust to various measures of distance including geographic distance, environmental distance, and genetic distance.

Interestingly, we found little to no signal of local adaptation in amoena populations. We hypothesize that this difference in the degree of adaptation within pilosa and amoena species could reflect either difference in migration or in selection. High migration between populations of amoena could cause homogenization of genetic variation across the range and swamping of locally adapted alleles. This is unlikely to explain the difference in pattern between species since the range of genetic distances ($F_{ST}$) represented in our experiment were similar for amoena (0–0.44) and pilosa (0–0.46) and for a given distance between populations, $F_{ST}$ is actually higher for amoena than pilosa (Goulet-Scott et al.[27]; Fig. 4B). In contrast, the range of environmental distances (based on a PCA of environmental variables) represented in our experiment was significantly less for amoena (0–3.48) than for pilosa (0–5.53). Therefore, pilosa populations may face stronger selection throughout their range to adapt to local ecological conditions.

By characterizing adaptive divergence between species and local adaptation within a species, we can compare and contrast how natural selection generates diversity across these scales. We quantify diversity in leaf morphological and physiological traits across species and find that different axes of diversity predict fitness between species versus within species. These three species of *Phlox* grow in close geographic proximity, share pollinators, and have similar flower shapes, sizes, colors, and timing. Therefore, their major phenotypic axis of diversity is in vegetative traits such as leaf morphology. Pilosa plants have long narrow leaves whereas amoena plants have shorter, wider leaves. Both species show extensive variation in the overall size and mass of the leaf.

Our results demonstrate that the major leaf trait differences between species strongly predict fitness variation across species in our common garden experiments. Plants with wider, shorter leaves do better in the amoena garden and plants with longer narrower leaves do better in the pilosa garden. It is perhaps unsurprising, that the traits that phenotypically differentiate species also predict fitness differences across the species' habitats. We have highlighted the link between key traits that define and differentiate closely related species, and fitness differences between species in their respective habitats.

This axis of phenotypic variation differentiating species (PC1) does not predict fitness variation within a species; instead, orthogonal trait variation (PC2) predicts within-species relative success. We found evidence of local adaptation across multiple proxies of fitness in pilosa that is predicted by a suite of leaf traits. Importantly, the strongest selection gradients within species are different in strength and direction from those inferred across species.

The observation that adaptation within and between species operates along different axes of selection might seem surprising given the perspective of a continuum of divergence between locally adapted

populations and ecologically isolated species. The ecological speciation hypothesis suggests that populations within a species diverge ecologically until those populations evolve sufficient reproductive isolation and become distinct species. This hypothesis has largely been evaluated by documenting a correlation across many pairs of lineages between ecological divergence and genetic divergence or reproductive isolation[1]. Here, we have demonstrated that the process of ecological adaptation is multidimensional: if ecological divergence along one axis leads to reproductive isolation and a signature of local adaptation between lineages, then local adaptation between populations within each lineage may persist or develop along other ecological axes. As has been articulated by others[23,25], the process of speciation is complex and not linear; similarly, the role of selection in driving divergence is also complex and multidimensional.

Further research is needed to determine if different ecological factors are more or less likely to drive between or within-species divergence. For example, adaptive divergence driven by ecological factors with discrete or step-like variation may be more likely to contribute to reproductive isolation between species due to the absence of intermediate habitat that could be suitable for hybrids[9,36]. In contrast, local adaptation to ecological factors that vary more continuously may be less likely to lead to reproductive isolation and therefore act among populations within species. Selective landscapes are clearly multifaceted; our study showcases this by demonstrating that different ecological forces generate divergence between closely related species than among populations within a species.

## Methods

### Ecological niche modeling

We used ecological niche modeling to assess environments occupied by our *Phlox* species. We combined coordinates from our field collections and occurrence data from the Global Biodiversity Information Facility (GBIF; https://www.gbif.org/) and the Southeast Regional Network of Expertise and Collections (SERNEC; https://sernecportal.org/portal/), including records within the native ranges that were identified to subspecies (*Phlox amoena* subsp. *amoena*, *Phlox pilosa* subsp. *deamii*, and *Phlox pilosa* subsp. *pilosa*). We thinned occurrences to one within 20 km using the R package 'spThin'[37] and retained 33 amoena, 87 pilosa, and only 5 deamii (Supplementary Data Table 2). We could not perform ecological niche modeling analyses for deamii due to low occurrences.

We extracted bioclimatic variables from the WorldClim dataset (https://www.worldclim.org/data/bioclim.html) and soil composition and chemistry variables from the Unified North America Soil Map (https://daac.ornl.gov/NACP/guides/NACP_MsTMIP_Unified_NA_Soil_Map.html) at each occurrence location for amoena and pilosa. We reduced collinearity between variables to retain 11 variables with correlation coefficients <0.8 (Supplementary Data Table 1). With these variables, we constructed Maxent ecological niche models for amoena and pilosa using the R package 'dismo'[38] following established protocols[39,40]. Model performance was evaluated using a repeated cross-fold approach in which 90% of the data were sampled to train a Maxent ENM before testing the model with the remaining 10% of the occurrence points. For both amoena (median testing AUC = 0.942) and pilosa (median testing AUC = 0.889), we were able to construct robust niche models (Fig. S1).

We performed a principal components analysis based on correlations on all environmental and soil variables used in our niche models. We assessed if the niches of the two species differed by comparing the empirical differences between species in median and breadth (difference between 5th and 95th percentile) along PC1 and PC2 to a null distribution defined by bootstrap resampling 1000 times the pooled and randomly reassigned occurrence points across both

species[39,41,42] (Fig. S1). This PCA was later used to calculate environmental distances between populations.

### Plant propagation

We propagated collections of 122 genotypes of *Phlox amoena amoena* (eight populations), 125 genotypes of *Phlox pilosa pilosa* (nine populations), and 37 genotypes of *Phlox pilosa deamii* (three populations) from throughout their native ranges for our common garden experiment (Supplementary Data Table 5). Wild plants were collected as cuttings of vegetative shoots and rooted and grown in the greenhouse facilities at the Arnold Arboretum of Harvard University. After growing for nine months replicate cuttings, each four inches in length, were taken from vegetative shoots on each plant and rooted and grown in fine potting media for one month before being transplanted into experimental gardens. To increase the sample size for deamii, we included two individuals per genotype in each garden, while only one individual per genotype was planted in each garden for the other two taxa.

### Experimental gardens and fitness measurements

We established three experimental gardens adjacent to one native population of each taxon (Supplementary Data Table 6). Each garden site contained four cleared plots into which the 321 plants were assigned a randomized position. Clonal cuttings from the greenhouse were planted in their assigned position, which was marked by an aluminum tag. Each plot was protected from large herbivores by PVC and chicken wire cages for one year after planting. The gardens were watered immediately after planting and then weekly for a month at which point supplemental watering stopped.

We monitored fitness-related traits in the gardens over the course of three growing seasons between planting in April 2018 and final data collection in September 2020. Survivorship across all three gardens in the first year was high (92.5% in amoena garden, 91.9% in deamii garden, 92.8% in pilosa garden, 92.4% total). At the end of the first winter in early 2019, we removed the wire cages and returned regularly to record traits throughout spring and summer. We recorded damage from large vertebrate herbivores as a binary trait (0 = herbivore damage, 1 = no herbivore damage). We counted the total number of open flowers on each plant on a weekly basis from the beginning of April through the beginning of June 2019. Flowers on these taxa remain open and fresh for about one week, so our timing minimized double counting or missing flowers. We counted the total number of fruits set by each plant including both mature fruits that remained on the plant as well as open calyces where fruits had already shattered. In October 2019, we harvested all aboveground biomass for each plant, leaving root systems and the stem at the base of each plant intact consistent with the annual aboveground die-back that these taxa experience each winter. We dried this tissue in a drying oven at 60 °C for 48 h before measuring the mass with an electronic scale. Due to the COVID-19 pandemic, we were not able to return to the gardens again until September 2020 when we recorded final survival.

### Between species adaptive divergence analyses

To test the hypothesis of adaptive differentiation between taxa, we used a generalized linear mixed model (GLMM) approach implemented in the R package 'lme4'[43,44]. For each fitness-related trait measured in the gardens, we modeled trait value with fixed effects of taxon, garden, and taxon-by-garden interaction and a random effect of genotype nested within the population. Each genotype occurred at least once as a clonal replicate in each garden. For herbivory and survival, we used a binomial link function in our models, while for a number of flowers and fruits, we used a Poisson link function. For biomass, we transformed the raw data by taking the natural logarithm and modeled this trait using a linear mixed model. After fitting each

model, we evaluated them using ANOVA as implemented in the R package 'car'[45].

The adaptive divergence between the species is expected to result in a significant taxon-by-garden interaction effect. Specifically, we predict the local taxon to outperform the two foreign taxa in its home garden (local vs. foreign comparisons) and/or for each taxon to perform better in its home garden than in the other two habitats (home vs. away comparison)[7]. To test these predictions we performed post-hoc contrasts using Tukey's Test as implemented in the R package 'multcomp'[46].

## Within-species local adaptation analyses

We implemented a provenance trial analysis to test for local adaptation and thus expected a negative correlation between a plant's performance and the distance between its source and the experimental garden in which it was measured. We calculated the distance between the experimental garden and the source population in three ways: geographic, genetic, and environmental (Supplementary Data Table 3). We calculated geographic distance with the longitude/latitude of each population's wild collection site and each experimental garden using the Haversine formula as implemented in the R package 'geosphere'[40]. We calculated the genetic distance as $F_{ST}$ between each wild source population and an intraspecific population adjacent to each experimental garden site. DNA sequencing and $F_{ST}$ calculations among these populations are detailed and reported in Goulet-Scott et al.[27]. Briefly, five individuals from each wild population were sequenced using double digest restriction-site associated DNA sequencing (ddRADseq), and all pairwise Weir-Cockerham $F_{ST}$ values between populations were calculated using VCFtools[47,48]. Finally, we calculated environmental distance as the Euclidean distance between each population's wild collection site and each experimental garden site in PC1 vs. PC2 space of the environmental PCA that accompanied the ecological niche modeling detailed above.

To quantify the contribution of the source population to the fitness of each clone in the experimental gardens, we used a GLMM. For each species, we modeled fitness trait value with a random effect of population nested within the garden, using the same link functions for each trait as described previously. These models yielded "population random effects" for each garden that estimated the average effect on the fitness trait value in that garden attributable to being from a given population. To test for local adaptation, we regressed population random effects for each trait/taxon combination against each measure of distance using linear models as implemented in base R[49]. For each linear model, we recorded the coefficient associated with the distance predictor, the coefficient of determination ($R^2$), and the associated *p*-value.

## Between and within-species trait selection analyses

Finally, we evaluated patterns of selection by determining how morphological and physiological trait variation predicted fitness both between and within species. We measured a standard suite of morphological and physiological traits on a clonal replicate of each experimental individual from the common garden and grown in the Arnold Arboretum greenhouse. These trait measurements required destructive sampling and were therefore not able to be measured on the plants growing in the field without compromising the experiment. From each plant, the most recently fully expanded leaf was collected, and the following measurements were taken: fresh mass, relative chlorophyll content using an atLeaf chlorophyll meter (FT Green, Wilmington, DE, USA), and dry mass. Each fresh leaf was scanned and we used ImageJ to measure leaf length, width, and area. We calculated specific leaf area (SLA) as area ($cm^2$) divided by dry mass (g). We summarized variation in leaf traits by performing principal component analysis (PCA) on leaf length, width, length/width ratio, area, relative chlorophyll content, and SLA using the correlation matrix. Together

the first two principal components described over 70% of the phenotypic variation and were thus used in subsequent analyses (PC1 = 45.0% of variation explained, PC2 = 27.8%). We used a linear model in R to determine the extent to which species identity explains variation on PC1 and PC2.

To confirm that the trait variation we measured is robust between the field and the greenhouse, we measured the same traits on individuals growing naturally in one of our source populations during the summer of 2018. This population contained both pilosa and amoena plants. We measured leaf length, leaf width, leaf area, and leaf dry mass, and calculated specific leaf area and leaf length/width ratio on 35 amoena plants and 37 pilosa plants growing in the natural population (population #729). We combined these field measurements with measurements taken from 29 plants sourced from this population grown in the greenhouse and used in the experimental gardens. We used an ANOVA model to determine the extent to which taxon (amoena vs. pilosa), location (greenhouse vs. field), and the interaction of taxon and location predicted leaf traits (Supplementary Data Table 11, Fig. S2). The trait best explaining PC1, length/width ratio, shows no difference between the field and greenhouse but a strong taxon effect which is consistent with all the greenhouse measurements. Leaf length, area, and width show significant taxon and location effects with field leaves being smaller than greenhouse leaves but the relationship between the taxon remains consistent across locations. We find a significant interaction between taxon and location for specific leaf area and leaf width. For leaf width, we find that the effect of being grown in the greenhouse (wider leaves) is slightly more for ameona than for pilosa but the rank order of the taxa remains the same across environments. In the case of specific leaf areas, we find that neither amoena nor pilosa shows significant differences between field and greenhouse-grown measurements and there is no overall effect of taxon or location. These results give us confidence that our greenhouse-based measurements are consistent with the relative variation measured between individuals growing in the field.

Because we were interested in understanding fitness variation both within and between species we focused our analyses on plants in the pilosa garden and the three fitness traits that showed both adaptive divergence between species and local adaptation within pilosa (flower number, fruit number, and biomass). For these analyses, fitness traits were normalized around the mean, and PC axes were z-transformed with a mean of 0 and a standard deviation of 1. First, we implemented two linear models in R, one for each of the first two PCs, to ask how PC of trait variation, taxon identity, and the interaction between these two main effects predicted fitness trait variation across all three species. Second, we implemented a series of simple linear models in R to specifically ask how PC1 and PC2 predicted fitness variation in four data sets: all species combined, only pilosa, only amoena, and only deamii. By comparing the results of these models for each fitness trait we assess whether the same dimension(s) of leaf trait variation predicted fitness within a species versus across all species together.

Principal components can be hard to interpret biologically, especially with regard to the impact of fitness. Therefore, we used the eigenvectors from our leaf trait PC and the selection gradients on the PC scores to reconstitute selection gradients onto the traits. This method is described in detail by Chong et al.[35]. In brief, we created a matrix of eigenvectors for each leaf trait and the first three PCs from our leaf trait PCA (referred to as E in formula (1) of Chong et al.; Supplementary Data Table 7). We generated a vector of selection gradients (referred to as A in formula (1) of Chong et al.; Supplementary Data Table 11) for the first three PC scores using both the full species dataset from the pilosa garden and only the pilosa individuals from the pilosa garden. We generated this vector for each of the three fitness traits (number of fruits, number of flowers, and biomass) that show evidence of selection both across species and within pilosa. The product of this matrix of eigenvectors and vector of selection coefficients is a vector

of reconstituted selection gradients for each leaf trait in the original PCA (Fig. 5H).

### Reporting summary

Further information on research design is available in the Nature Portfolio Reporting Summary linked to this article.

## Data availability

The data collected in this study are available on Dryad at DOI: 10.5061/dryad.gxd2547sx. All collated data and summarized data are available as Supplementary information and data. Raw sequence data used in this project are available on the NCBI sequence read archive: PR-JNA701424 (https://www.ncbi.nlm.nih.gov/bioproject/701424)

## Code availability

Source code is available at https://github.com/PhloxHopkins/PhloxFieldAdaptation

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

## Acknowledgements

We are grateful to Dan Hartl, Missy Holbrook, James Mallet, Grace Burgin, Austin Garner, Samridhi Chaturvedi, Jake Grossman, and Antonio Serrato-Capuchina for their helpful discussions. We would also like to thank Abraham Cone, Andrew Goulet, Julian Campbell, Jewel Cotton, Neville Crawford, Carolyn Ferguson, Elizabeth Raikes, and Richard Carter (Valdosta State University Herbarium) for assistance with collecting plants used in this experiment, the Arnold Arboretum greenhouse staff for plant care, and Hancock Biological Station (Murray State University). Finally, we thank the US Forest Service (Land between the lakes permit LBL17170), Paris Bailey, and Jewel Cotton for permission to plant experimental gardens on their land. Plants for this experiment were collected with the following permits: USDA Forest Service # 005277 & SCDNR #SU-12-2017. This study was funded by the National Science Foundation (DEB-1844906 to R.H.), and a Rosemary Grant Advanced Award from the Society for the Study of Evolution to B.G.-S. M.B. was supported by a National Science Foundation Postdoctoral Research Fellowship in Biology under the Plant Genome Research Program.

## Author contributions

B.G.-S. and R.H. designed the experiment; B.G.-S., M.F., A.B., C.H., and R.H. contributed to data collection; B.G.-S., M.B., and R.H. contributed to data analysis, B.G.-S. and R.H. wrote the manuscript with all authors contributing to editing and revising.

## Competing interests

The authors declare no competing interests.
