## [Peer Review File · Nature Communications]

A multi-dimensional selective landscape drives adaptive divergence between and within closely related Phlox speciesReviewer #1 (Remarks to the Author):

The authors have produced a well-written manuscript from an impressive study that tackles a compelling question: Do the axes of selection that favor adaptive divergence between species align with those that favor local adaptation within a species? This question is both timely and significant.

The manuscript thoroughly investigates the adaptive divergence between closely related *Phlox* species and local adaptation within these species. The study employs ecological niche modelling, principal component analysis, and reciprocal transplant experiments to elucidate the complex interplay of natural selection at both micro- and macro-evolutionary scales. The Discussion section adds depth by exploring the multi-dimensionality of selective forces and the role of ecological factors in shaping these landscapes.

GENERAL STRENGTHS

Using ecological niche models, PCA, and reciprocal transplant experiments provides a robust framework for studying adaptive divergence and local adaptation. This is an impressive study.

The paper excellently captures the complexity of natural selection, emphasizing that divergence within and between species can occur along orthogonal axes of selection. This nuanced approach is commendable.

The manuscript significantly contributes by showing how different traits predict fitness within and between species, adding to our understanding of adaptive divergence mechanisms.

GENERAL CRITIQUE

While the paper discusses ecological factors' role in adaptive divergence and local adaptation, it would benefit from a deeper exploration of these complexities. Incorporating more empirical or theoretical models could provide additional insights.

The paper provides a detailed account of local adaptation, particularly in ecological gradients. Future work might explore how these gradients interact with factors like gene flow and mutation rates to shape local adaptation patterns.

The paper concludes by highlighting the non-linear and complex nature of speciation. It would be beneficial for the authors to expand on how their findings fit into broader theories of speciation, perhaps by referencing other works that also view speciation as a multi-dimensional process (e.g., <https://doi.org/10.1093/evolut/qpac004>)

Given the focus on trait variation and its correlation with fitness, future studies might explore the genetic mechanisms underlying these variations. Also, relating this work to previous work in *Phlox* would help the reader follow the story of speciation in this fascinating system.

METHODS STRENGTHS

The Methods section is well-structured and provides a comprehensive overview of the various techniques and analyses employed in the study. The authors have used a multi-pronged approach that combines Maxent models and principal components analysis (PCA) to assess the ecological niches of *Phlox amoena* and *Phlox pilosa*. Using WorldClim and Unified North America Soil Map datasets adds credibility to the environmental variables. The authors reveal detailed procedures for plant propagation, including the use of clonal cuttings and greenhouse facilities, which are well-documented. The inclusion of multiple genotypes from different populations adds rigor to the study. The establishment of experimental gardens and the meticulous recording of fitness-related traits over three growing seasons (!!!) provide a robust dataset for analysis.

Using generalized linear mixed models (GLMM) and provenance trial analysis adds statistical rigor to the study. GLMM and ANOVA are appropriate and well-justified for adaptive divergence analyses and linear models for trait selection analyses. The paper also employs post-hoc contrasts using Tukey's Test, adding another layer of statistical rigor. Using various R packages for data handling, statistical analyses, and model evaluation is commendable. The authors have also taken steps to

validate their greenhouse-based measurements with field data, which adds to the study's reliability.

METHODS CRITIQUE

The paper acknowledges the limitation of low occurrences for *Phlox pilosa* subsp. *deamii* in ecological niche modeling. Future work could explore ways to overcome this limitation by using other data types or statistical techniques that can handle low sample sizes -- add a note on this in the discussion. The impact of the COVID-19 pandemic on data collection is noted but needs to be elaborated upon. Discussing how this may have affected the study's results or interpretations would be beneficial. *** While the paper employs multiple statistical tests, it would be prudent to apply corrections for multiple comparisons to ensure the robustness of the findings.

POINT OF CONTENTION

Reinforcement speciation is common in *Phlox* and has been extensively documented by the authors. In reinforcement speciation, the primary objective of natural selection is to reduce interspecific gene flow through ecological or reproductive character displacement. This is achieved by evolving mechanisms that prevent hybridization, and not necessarily by driving adaptation to different ecological niches. This is the domain of ecological speciation, where adaptation to different environments leads to the indirect evolution of reproductive isolation between species. Overall, reinforcement focuses mainly on the evolution of reproductive isolation mechanisms in response to hybridization costs, not due to local adaptation within species. One possible explanation for decoupling within and between species divergence is that reinforcement drives it. I would have liked to see a deeper consideration of this point because most other studies where the within-between axes of divergence hypothesis can be tested will not have this complication.

The work of Dolph Schluter and others has tackled cases of within and between ecological divergence (e.g., <https://www.pnas.org/doi/10.1073/pnas.0508653103>) and the ecological speciation hypothesis, for which there is ample evidence, directly posits an alignment between the axis of divergence within and between species. Further, a recent paper on multiple axes of divergence (doi above) during speciation has delved into the issue presented in this paper more broadly.

Also, a relatively old paper on "Genomics and the origin of species" explicitly considered the decoupling of within and between reproductive isolation and ecological adaptation in box two.

In summary, while reinforcement and ecological divergence are mechanisms of adaptive divergence, they operate under different conditions and result in different evolutionary changes. Reinforcement is more about "fine-tuning" species to avoid the costs of hybridization, while ecological divergence is more about "retooling" populations to thrive in different ecological niches.

***In an extreme case, the authors have an excellent opportunity to frame their study regarding reinforcement, which they should have addressed more strongly, regardless. They could explore the congruence between ecological divergence between and within species during *Phlox* reinforcement, where some of the known ecological factors driving reproductive isolation in sympatry are known to be ecological. To clarify my meaning, you could think that a new title for the paper could be: "The Multi-dimensional Selective Landscape of Speciation by Reinforcement in Closely Related *Phlox* Species."

CONCLUSION

The manuscript offers a comprehensive and nuanced view of adaptive divergence and local adaptation in *Phlox* species. It successfully integrates complex datasets to provide valuable insights into the multi-dimensional nature of natural selection. With some minor revisions, this paper will significantly contribute to the evolutionary biology field.

**** Appendix

Notes on Reinforcement and Local Adaptation

Comparison of the two processes and why the result from the authors follows from these premises:

Driver of Divergence: In reinforcement speciation, the driver is often the avoidance of producing fewer fit hybrids. In ecological divergence, the driver is local adaptation to different ecological conditions.

Gene Flow: In reinforcement, gene flow between species is reduced by evolving mechanisms to avoid hybridization. In ecological divergence, gene flow within a species between different populations is reduced due to local adaptations.

Geographic Overlap: Reinforcement speciation specifically occurs in sympatric areas where two species coexist. Ecological divergence often starts in allopatric settings where populations are geographically isolated but can also occur in sympatric or parapatric settings under certain conditions.

Outcome: Both mechanisms ultimately lead to speciation but through different pathways. Reinforcement strengthens pre-existing reproductive barriers, while ecological divergence may create new species adapted to different ecological niches.

Adaptive Traits: In reinforcement, the evolving traits are specifically related to reproductive isolation. In ecological divergence, the traits are related to survival and reproduction in a specific environment.

Time Scale: Both can occur on varying time scales, but reinforcement might happen more quickly if hybridisation costs are high, forcing rapid evolution of isolating mechanisms.

Genetic Architecture: The genetic changes involved in reinforcement often focus on a few key traits related to reproduction. In ecological divergence, the changes can be more widespread, affecting various aspects of physiology, behaviour, and morphology.

Reviewer #2 (Remarks to the Author):

This manuscript reports on a reciprocal common-garden study designed to explore patterns of divergence and adaptation in a set of 3 closely related Phlox species in the Eastern US. The main strength and novelty of the study is the inclusion of both the species and population level. While patterns of local adaptation within species and patterns of species divergence are often studied in isolation, the current study integrates both. There are other systems where work has been done across levels, but to my knowledge not using repeated, reciprocal transplants of the kind used here.

The results are interesting, revealing patterns of apparent adaptation of each species to its habitat, as well as local adaptation within species. Unfortunately, understanding the implications of the exciting data and interesting results is seriously compromised by the current presentation of the study.

The main issue with the current version of the paper is lack of quantification and quantitative interpretation. The results are presented very qualitatively, with most statements focusing on the presence or absence of effect, rather than the size of that effect. An example is the presentation of regression relationships (e.g. local performance vs. distance) as "correlations", and local advantage as "significant" rather than e.g. reported as a percent advantage. This is problematic in a highly quantitative field such as evolutionary biology, where the evolutionary importance of any effect must be understood in units that are relevant to the process under study (here, evolutionary divergence of populations and species).

One of the main claims of the papers is that axes of selection differ (indeed, are orthogonal)

between the species and population level, i.e. that different and perhaps even independent patterns of selection have driven population and species divergence. If it was strictly true that axes of selection were completely independent, this could have wide-ranging implications. However, the evidence of such independence appears to be that the leading principal component of trait data is detectably (read statistically significantly) associated with performance variation across species, while the second principal component is detectably associated with divergence. This illustrates the issue with the current presentation: without knowledge of the actual parameter estimates it is almost impossible to know how similar or dissimilar the patterns of selection are.

A related issue in this part of the analysis is that it involves phenotypic-selection analyses, for which there is a well-defined theoretical framework outlined by Lande and Arnold in 1983. In this framework the response variable is relative fitness (individual fitness divided by population mean fitness), which yields regression slopes (selection gradients) interpretable as the proportional change in fitness per unit trait change (where a unit is usually either a standard deviation or a trait mean). In the current analysis the predictor variables (PC axes) are variance-scaled but the response variable is absolute rather than relative fitness, and thus no selection gradients are reported (in fact, the slopes are not reported or interpreted, at least not in the main text). The presentation of these analyses should involve quantitative interpretation of the selection gradients (including comparison to typical values from other studies and meta-analyses) and their similarity between the different levels. The use of PC axes complicates this further. One option could perhaps be to use the approach of Chong et al. 2018 *EvolLett* to project the selection gradients on the PCs back onto the original traits and then compare those between the population and species level (by simply correlating them or computing e.g. the angle between them). Alternatively, all traits could be included in the analysis (barring possible multicollinearity issues).

Some other comments

When I first read the manuscript, I was highly surprised to read a paper on divergence in *Phlox* without (nearly) any mention of pollination and floral traits. This surprised me as the Polemoniaceae was a key study system for Grant in his work on pollinator-mediated divergence of populations and species, as well as later work by Levin and others on *Phlox* in particular. The authors surely know this literature well, but it not clear why they have chosen to "ignore" it in this study. Line 305-309 hints at less differentiation in pollinators and flowers in these taxa, but this needs to be clarified from the start.

The Discussion, especially the first pages, reads as a long summary of the results with no references or wider discussion. I think the Discussion would benefit from reducing the study-specific details and increasing general points and reference to other systems.

Line 52-56 and later Line 235: I agree there are few studies considering patterns of adaptation across both populations and species, but there are some. One example is Scott Armbruster's work with *Dalechampia* (e.g. Armbruster 1988 *Ecology*, and more recent ones e.g. Opedal et al. 2023 *AJB*), another is perhaps some of John Thompson's work with *Lithophragma* (e.g. Thompson et al. 2013 *PNAS*, 2017 *AmNat*).

Line 63: Not clear from this formulation what the role of genetic correlations are in this context. The cited references are to studies focusing on constraints as manifest as divergence in directions of high genetic variation/evolvability, there are several recent studies taking that view one step further (e.g. Houle et al. 2017 *Nature*, McGlothlin et al. 2018 *AmNat*, Opedal et al. 2023 *PNAS*). That being said, the manuscript focuses on selection, not on evolvability/genetic architecture/constraints.

Line 237-239: What is the evidence of "orthogonal" selection?

Line 274-276: But how does this relate to the decay in performance with distance from garden?

Line 471-491: To me this seems unnecessarily detailed. Obtaining trait measurements for each genotype in the greenhouse seems to me a strength rather than a weakness of the study, as this should eliminate some small-scale environmental variation in the field that could cause spurious trait-fitness correlations. Thus, I would present the decision to measure traits in the greenhouse as

a strength!

We thank the editor and reviewers for their thoughtful and helpful reviews. We are grateful for the comments and feel the manuscript is significantly improved. We are particularly appreciative for the reviewers' suggestion for additional analyses and points needing clarification. Please see below for our point-by-point responses.

REVIEWER COMMENTS

The authors have produced a well-written manuscript from an impressive study that tackles a compelling question: Do the axes of selection that favor adaptive divergence between species align with those that favor local adaptation within a species? This question is both timely and significant.

The manuscript thoroughly investigates the adaptive divergence between closely related Phlox species and local adaptation within these species. The study employs ecological niche modelling, principal component analysis, and reciprocal transplant experiments to elucidate the complex interplay of natural selection at both micro- and macro-evolutionary scales. The Discussion section adds depth by exploring the multi-dimensionality of selective forces and the role of ecological factors in shaping these landscapes.

GENERAL STRENGTHS

Using ecological niche models, PCA, and reciprocal transplant experiments provides a robust framework for studying adaptive divergence and local adaptation. This is an impressive study.

The paper excellently captures the complexity of natural selection, emphasizing that divergence within and between species can occur along orthogonal axes of selection. This nuanced approach is commendable.

The manuscript significantly contributes by showing how different traits predict fitness within and between species, adding to our understanding of adaptive divergence mechanisms.

We thank the reviewer for their overall encouraging and positive assessment of our manuscript.

GENERAL CRITIQUE

While the paper discusses ecological factors' role in adaptive divergence and local adaptation, it would benefit from a deeper exploration of these complexities. Incorporating more empirical or theoretical models could provide additional insights.

We thank the reviewer for this thought and agree that this area of research would benefit from deeper research. Beyond the verbal models surrounding hypotheses about ecological speciation (as discussed on lines 70-78), we are unaware of other theoretical models that would be applicable to this situation of selection across scales of populations and species.

The paper provides a detailed account of local adaptation, particularly in ecological gradients. Future work might explore how these gradients interact with factors like gene flow and mutation rates to shape local adaptation patterns.

We appreciate the reviewers bringing this up and find it an incredibly fascinating future direction. The fact that across populations, adaptations are maintained despite gene flow and between species, gene flow is prevented by other adaptations provides a fascinating arena to

study gene flow and mutation rates. Although, addressing this empirically is beyond the scope of this work.

The paper concludes by highlighting the non-linear and complex nature of speciation. It would be beneficial for the authors to expand on how their findings fit into broader theories of speciation, perhaps by referencing other works that also view speciation as a multi-dimensional process ([doi10.1093/evolut/qpac004](https://doi.org/10.1093/evolut/qpac004)).

We thank the reviewer for bringing this paper to our attention. We have added the citation to our discussion of other models that think about the continuum(s) of speciation in our introduction (line 89,) as well as to our discussion of the complexity of speciation and continuums in our discussion (line 387). We note that the Bolnick et al perspective is distinct from what we found in our study. The Bolnick et al paper articulates how divergence between species and the speciation process may occur along many axes including phenotypic, ecological, genetic, and reproductive isolation. These axes can interact or be of varying degrees of importance leading to multidimensional speciation processes. Our findings are that the axis of ecological divergence (and phenotypic divergence) and local adaptation is not actually an axis on which closely related populations are at one end and distantly related species are at the other end. Instead, entirely different traits, selection pressures, and ecological factors are differentiating populations versus species.

Given the focus on trait variation and its correlation with fitness, future studies might explore the genetic mechanisms underlying these variations. Also, relating this work to previous work in *Phlox* would help the reader follow the story of speciation in this fascinating system.

We completely agree! We are excited to start investigating the traits underlying these patterns at a more mechanistic and genetic level.

METHODS STRENGTHS

The Methods section is well-structured and provides a comprehensive overview of the various techniques and analyses employed in the study. The authors have used a multi-pronged approach that combines Maxent models and principal components analysis (PCA) to assess the ecological niches of *Phlox amoena* and *Phlox pilosa*. Using WorldClim and Unified North America Soil Map datasets adds credibility to the environmental variables. The authors reveal detailed procedures for plant propagation, including the use of clonal cuttings and greenhouse facilities, which are well-documented. The inclusion of multiple genotypes from different populations adds rigor to the study. The establishment of experimental gardens and the meticulous recording of fitness-related traits over three growing seasons (!!!) provide a robust dataset for analysis.

Using generalized linear mixed models (GLMM) and provenance trial analysis adds statistical rigor to the study. GLMM and ANOVA are appropriate and well-justified for adaptive divergence analyses and linear models for trait selection analyses. The paper also employs post-hoc contrasts using Tukey's Test, adding another layer of statistical rigor. Using various R packages for data handling, statistical analyses, and model evaluation is commendable. The authors have also taken steps to validate their greenhouse-based measurements with field data,

which adds to the study's reliability.

We thank the reviewer for noting our rigor and experimental thoroughness.

METHODS CRITIQUE

The paper acknowledges the limitation of low occurrences for *Phlox pilosa* subsp. *deamii* in ecological niche modeling. Future work could explore ways to overcome this limitation by using other data types or statistical techniques that can handle low sample sizes -- add a note on this in the discussion. The impact of the COVID-19 pandemic on data collection is noted but needs to be elaborated upon. Discussing how this may have affected the study's results or interpretations would be beneficial. *** While the paper employs multiple statistical tests, it would be prudent to apply corrections for multiple comparisons to ensure the robustness of the findings.

POINT OF CONTENTION

Reinforcement speciation is common in *Phlox* and has been extensively documented by the authors. In reinforcement speciation, the primary objective of natural selection is to reduce interspecific gene flow through ecological or reproductive character displacement. This is achieved by evolving mechanisms that prevent hybridization, and not necessarily by driving adaptation to different ecological niches. This is the domain of ecological speciation, where adaptation to different environments leads to the indirect evolution of reproductive isolation between species. Overall, reinforcement focuses mainly on the evolution of reproductive isolation mechanisms in response to hybridization costs, not due to local adaptation within species. One possible explanation for decoupling within and between species divergence is that reinforcement drives it. I would have liked to see a deeper consideration of this point because most other studies where the within-between axes of divergence hypothesis can be tested will not have this complication.

We appreciate the author's thoughts on reinforcement and how alternative mechanisms might be involved in the divergence we see. We respectfully disagree and think there is no evidence of reinforcement in this system. As pointed out by the reviewer, reinforcement involves divergence within a species in response to costly hybridization between species. This divergence within a species specifically decreases hybridization between species and thus impacts traits involved in pre-zygotic reproductive isolation. In contrast, adaptation both across populations within species and between species is in response to natural selection favoring traits that increase survival or reproductive success (fitness). As explained below, we see evidence of adaptive divergence and not divergence impacting hybridization through reinforcement.

If our patterns of within species divergence were due to reinforcement, then we would expect 1) divergence to be in traits that impact hybridization rates such as floral traits, or pollen-pistil interactions; 2) Divergence between populations of *pilosa* to correspond to proximity to, frequency of, or interactions with amoena. Although other species of *Phlox* show divergence in floral color that has been attributed to changes in mating and hybridization, the species in this study have nearly indistinguishable flowers. We see little to no evidence of pre-mating reproductive isolation. In fact, in the common gardens we created for this study, we observed pollinators moving between species just as much as within species (unpublished data).

Furthermore, we can generate hybrids in the lab between these species using controlled crosses and see no variation in success depending on geographic location. Finally, we see no evidence that divergence between populations of pilosa corresponds to proximity to or interactions with the amoena species. We specifically sampled the two species across the broadly overlapping portion of their ranges and thus the divergence we see across populations within a species does not correspond to geographic proximity between species.

Instead, we found that survival, growth, and fruit set traits were differentiated across populations and species as would be expected for divergence due to natural selection for local adaptation. We also found that variation in these traits were correlated with the environment they were grown in and not the presence/absence of conspecifics. The differences in fitness both within and between species was highly predicted by environmental differences between the garden they were grown in and their source location.

The work of Dolph Schluter and others has tackled cases of within and between ecological divergence (e.g., <https://www.pnas.org/doi/10.1073/pnas.0508653103>) and the ecological speciation hypothesis, for which there is ample evidence, directly posits an alignment between the axis of divergence within and between species. Further, a recent paper on multiple axes of divergence (doi above) during speciation has delved into the issue presented in this paper more broadly.

Also, a relatively old paper on “Genomics and the origin of species” explicitly considered the decoupling of within and between reproductive isolation and ecological adaptation in box two.

We appreciated the suggested citations and have added them. We have also clarified how our experimental design has differed from other studies such as those mentioned. A number of studies have addressed the hypothesis of ecological speciation by looking at many pairs of lineages and comparing ecological divergence with genetic divergence or reproductive isolation. This presumes that the divergence occurs along a consistent ecological gradient. Instead, we test the same individuals for local adaptation across populations and between species to see how both processes can simultaneously occur and do not happen along the same continuum. We clarify this on lines 381-383.

In summary, while reinforcement and ecological divergence are mechanisms of adaptive divergence, they operate under different conditions and result in different evolutionary changes. Reinforcement is more about "fine-tuning" species to avoid the costs of hybridization, while ecological divergence is more about "retooling" populations to thrive in different ecological niches.

***In an extreme case, the authors have an excellent opportunity to frame their study regarding reinforcement, which they should have addressed more strongly, regardless. They could explore the congruence between ecological divergence between and within species during Phlox reinforcement, where some of the known ecological factors driving reproductive isolation in sympatry are known to be ecological. To clarify my meaning, you could think that a new title for the paper could be: “The Multi-dimensional Selective Landscape of Speciation by Reinforcement in Closely Related Phlox Species.”

We thank the reviewer for this suggestion and hope that our explanation clarifies why this study

measures adaptation driven by natural selection and not reinforcement.

CONCLUSION

The manuscript offers a comprehensive and nuanced view of adaptive divergence and local adaptation in Phlox species. It successfully integrates complex datasets to provide valuable insights into the multi-dimensional nature of natural selection. With some minor revisions, this paper will significantly contribute to the evolutionary biology field.

**** Appendix

Notes on Reinforcement and Local Adaptation

Comparison of the two processes and why the result from the authors follows from these premises:

Driver of Divergence: In reinforcement speciation, the driver is often the avoidance of producing fewer fit hybrids. In ecological divergence, the driver is local adaptation to different ecological conditions.

We entirely agree and think that figure 3 and 4 demonstrate that the divergence is due to different ecological conditions.

Gene Flow: In reinforcement, gene flow between species is reduced by evolving mechanisms to avoid hybridization. In ecological divergence, gene flow within a species between different populations is reduced due to local adaptations.

We find no evidence for variation in hybridization (or hybridization at all) but instead find significant evidence for immigrant inviability reducing the opportunity for species to reproduce as is consistent with ecological divergence.

Geographic Overlap: Reinforcement speciation specifically occurs in sympatric areas where two species coexist. Ecological divergence often starts in allopatric settings where populations are geographically isolated but can also occur in sympatric or parapatric settings under certain conditions.

We argue that ecological divergence can occur on many different scales – sometimes over large geographic differences such as between northern and southern populations and sometimes over microhabitat differences such as what differentiates the species at a fine geographic scale.

Outcome: Both mechanisms ultimately lead to speciation but through different pathways. Reinforcement strengthens pre-existing reproductive barriers, while ecological divergence may create new species adapted to different ecological niches.

Adaptive Traits: In reinforcement, the evolving traits are specifically related to reproductive isolation. In ecological divergence, the traits are related to survival and reproduction in a specific environment.

We find the traits we find associated with higher fitness are vegetative traits and not reproductive traits and thus believe they are associated with size and survival and not mate choice.

Time Scale: Both can occur on varying time scales, but reinforcement might happen more quickly if hybridisation costs are high, forcing rapid evolution of isolating mechanisms.

Genetic Architecture: The genetic changes involved in reinforcement often focus on a few key traits related to reproduction. In ecological divergence, the changes can be more widespread, affecting various aspects of physiology, behaviour, and morphology.

Reviewer #2 (Remarks to the Author):

This manuscript reports on a reciprocal common-garden study designed to explore patterns of divergence and adaptation in a set of 3 closely related *Phlox* species in the Eastern US. The main strength and novelty of the study is the inclusion of both the species and population level. While patterns of local adaptation within species and patterns of species divergence are often studied in isolation, the current study integrates both. There are other systems where work has been done across levels, but to my knowledge not using repeated, reciprocal transplants of the kind used here.

The results are interesting, revealing patterns of apparent adaptation of each species to its habitat, as well as local adaptation within species. Unfortunately, understanding the implications of the exciting data and interesting results is seriously compromised by the current presentation of the study.

The main issue with the current version of the paper is lack of quantification and quantitative interpretation. The results are presented very qualitatively, with most statements focusing on the presence or absence of effect, rather than the size of that effect. An example is the presentation of regression relationships (e.g. local performance vs. distance) as “correlations”, and local advantage as “significant” rather than e.g. reported as a percent advantage. This is problematic in a highly quantitative field such as evolutionary biology, where the evolutionary importance of any effect must be understood in units that are relevant to the process under study (here, evolutionary divergence of populations and species).

One of the main claims of the papers is that axes of selection differ (indeed, are orthogonal) between the species and population level, i.e. that different and perhaps even independent patterns of selection have driven population and species divergence. If it was strictly true that axes of selection were completely independent, this could have wide-ranging implications. However, the evidence of such independence appears to be that the leading principal component of trait data is detectably (read statistically significantly) associated with performance variation across species, while the second principal component is detectably associated with divergence.

This illustrates the issue with the current presentation: without knowledge of the actual parameter estimates it is almost impossible to know how similar or dissimilar the patterns of selection are.

A related issue in this part of the analysis is that it involves phenotypic-selection analyses, for which there is a well-defined theoretical framework outlined by Lande and Arnold in 1983. In this framework the response variable is relative fitness (individual fitness divided by population mean fitness), which yields regression slopes (selection gradients) interpretable as the proportional change in fitness per unit trait change (where a unit is usually either a standard deviation or a trait mean). In the current analysis the predictor variables (PC axes) are variance-scaled but the response variable is absolute rather than relative fitness, and thus no selection gradients are reported (in fact, the slopes are not reported or interpreted, at least not in the main text). The presentation of these analyses should involve quantitative interpretation of the selection gradients (including comparison to typical values from other studies and meta-analyses) and their similarity between the different levels. The use of PC axes complicates this further. One option could perhaps be to use the approach of Chong et al. 2018 *EvolLett* to project the selection gradients on the PCs back onto the original traits and then compare those between the population and species level (by simply correlating them or computing e.g. the angle between them). Alternatively, all traits could be included in the analysis (barring possible multicollinearity issues).

The review brings up an issue that we struggled with, and we appreciate the helpful suggestion. Including the reconstituted selection gradients of the original traits has greatly improved our manuscript. As noted, estimating selection on traits that are highly correlated is challenging. Because we were not concerned with the particulars of the traits, we chose to synthesize the phenotypes with a PCA. This can present problems of biological and statistical interpretation as noted by the reviewer. We performed the reviewer's suggested analysis (as described by Chong et al 2018) and find that it not only supports our original conclusions but clarifies our interpretation substantially. We have added a panel H to figure 5 showing the projected selection gradients for each trait and three fitness measures. This analysis was performed for all the species and for only the pilosa individuals. Comparing the vectors of selection gradients from these two datasets we find major differences in magnitude and direction of selection as would be expected if selection acts on different traits between versus within species. We have added this analysis to our results (lines 243-256) and methods (lines 563-574) and thank the reviewer for this very helpful suggestion.

Some other comments

When I first read the manuscript, I was highly surprised to read a paper on divergence in Phlox without (nearly) any mention of pollination and floral traits. This surprised me as the Polemoniaceae was a key study system for Grant in his work on pollinator-mediated divergence of populations and species, as well as later work by Levin and others on Phlox in particular. The authors surely know this literature well, but it not clear why they have chosen to "ignore" it in this study. Line 305-309 hints at less differentiation in pollinators and flowers in these taxa, but this needs to be clarified from the start.

We thank the reviewer for bringing this up and have clarified that these species have very similar floral morphologies, colors, timing, and sizes. Grant, Levin and our lab has done extensive work on floral variation in Phlox, these particular species have very little variation. We note this now on line 105-106 & 353-354.

The Discussion, especially the first pages, reads as a long summary of the results with no references or wider discussion. I think the Discussion would benefit from reducing the study-specific details and increasing general points and reference to other systems.

We have cut and rearranged the discussion to accommodate this suggestion.

Line 52-56 and later Line 235: I agree there are few studies considering patterns of adaptation across both populations and species, but there are some. One example is Scott Armbruster's work with Dalechampia (e.g. Armbruster 1988 Ecology, and more recent ones e.g. Opedal et al. 2023 AJB), another is perhaps some of John Thompson's work with Lithophragma (e.g. Thompson et al. 2013 PNAS, 2017 AmNat).

Line 63: Not clear from this formulation what the role of genetic correlations are in this context. The cited references are to studies focusing on constraints as manifest as divergence in directions of high genetic variation/evolvability, there are several recent studies taking that view one step further (e.g. Houle et al. 2017 Nature, McGlothlin et al. 2018 AmNat, Opedal et al. 2023 PNAS). That being said, the manuscript focuses on selection, not on evolvability/genetic architecture/constraints.

We appreciate the request for clarity on this point. Our point was simply that divergence in response to selection was very common but not inevitable due to other factors such as genetic constraint and competing selection pressures. We have adjusted our wording to hopefully clarify our intention here (line 61-62).

Line 237-239: What is the evidence of "orthogonal" selection?

We have clarified our meaning with this line and removed the word "orthogonal".

Line 274-276: But how does this relate to the decay in performance with distance from garden? We now show the differences in selection gradients when comparing performance of individuals from within pilosa (from different distances) versus individuals across all the species.

Line 471-491: To me this seems unnecessarily detailed. Obtaining trait measurements for each genotype in the greenhouse seems to me a strength rather than a weakness of the study, as this should eliminate some small-scale environmental variation in the field that could cause spurious trait-fitness correlations. Thus, I would present the decision to measure traits in the greenhouse as a strength!

We are grateful that the reviewer sees the strength in our experimental approach and have decided to keep this analysis in the manuscript in an effort to be as transparent with our methods and robust with our analyses.

Reviewer #1 (Remarks to the Author):

Thanks for addressing all the comments from both of us. It is a much-improved manuscript.

Reviewer #2 (Remarks to the Author):

This is my second time reviewing this manuscript. The authors have made a number of changes from the last version, including implementing one of my own suggestions from the last round, computing and reporting selection coefficients for PC axes projected back into the original trait space. I think the manuscript has improved, but I still believe there is a lack of quantification and quantitative interpretation of the results that makes it difficult to judge the meaning of the otherwise interesting results. In the following I have tried to give some very concrete examples of where I think quantification is warranted but currently missing.

For example, the section "Adaptive divergence between taxa" in the results completely lack quantification and quantitative interpretation. One example:

"We quantified five fitness-related traits: herbivory, fruit number, flower number, biomass, and survival and find that a significant taxon-by-garden interaction predicts trait values, indicating that the relative success of the species depends on the garden in which they are grown (Fig. 3; Table 1). Adaptive divergence is evidenced by either the local species having higher fitness than the foreign species in the local species' garden, or by a focal species having highest fitness in its home garden compared to all other away gardens."

Here I would want, as a reader, to know how many % higher fitness the local species has compared to foreign species, or by how many % the local species did better in its own habitat. In the first sentence, I would present biology first and statistics later: "We quantified five fitness-related traits: herbivory, fruit number, flower number, biomass, and survival and find that the relative success of the species depends on the garden in which they are grown, as indicated by statistical support for a taxon-by-garden interaction (Fig. 3; Table 1)"

Another example:

"amoena had higher survival and experienced significantly less herbivory than both pilosa and deamii, and produced more fruits than pilosa plants. In the deamii habitat, deamii had significantly higher survival than pilosa. In the pilosa habitat, pilosa plants produced significantly more fruits, and survived at a higher rate than both amoena and deamii plants. "

What is "significantly less"? What is "more fruits"? "A higher rate"? Please tell us what the differences were, not only if they were statistically supported or not.

Similarly, some examples from the section "Local adaptation within species":

"Local adaptation was evidenced by a negative relationship between the estimated population effect on fitness and distance of the population from the common garden."

Please present slope, standard error, and units. "Reproductive performance in terms of fruit set decreased by x% per km distance between the garden site and the natural population (slope = $x \pm y$ fruits/km)"

"Final biomass in pilosa also shows a strong negative correlation with geographic distance"

Here, something like "Biomass decreased by x.xx g per km increase in distance ()"

Related issues occur in the section "Selection between and within species", and here there is also substantial ambiguity in how the analyses were done.

"Due to collinearity between traits we summarized phenotypic variation using a principal components analysis (Fig. 5)."

Please state whether the PCA was performed on standardized data (I think it was).

"We investigated how leaf trait variation (PC1 and PC2) explained variation in fitness traits (fruit set, flower set, and biomass) in the pilosa garden"

Does this mean a formal phenotypic-selection analysis was done (regressing relative fitness on the traits)? There are contradictory statements here and in the Method. Be explicit.

"For PC2 the interaction term in our model is significant indicating that the relationship between PC2 and fitness traits varied across species (Table S8)."

Selection varied between species, as indicated by support for the interaction term?

"The results reveal that fitness differences between species is due to strong selection acting on leaf length, and leaf length/width ratio (leaf shape), whereas within pilosa there is strong selection acting on specific leaf area, leaf area, and chlorophyll content."

What is "strong selection"? First, we need to know how selection was quantified. I think variance-standardized selection gradients, but not sure (was the response variable relative or absolute fitness? If the values are indeed variance-standardized selection gradients, the interpretation is % change in fitness per SD change in the trait. Quantify and judge strength based on the literature (see e.g. Hereford et al. 2004 Evolution for a thorough discussion about interpretation of the strength of selection).

To say something about the difference in selection between levels, one method could be to calculate the angle between the selection gradients (if the reported values are formal selection gradients, see above). For the first fitness component (#Flowers), this could be done as follows in R:

```
v1 = c(-0.09, -0.02, 0.12, 0.12, -0.03, 0.16)
v2 = c(-0.25, 0.16, -0.01, 0.11, 0.07, 0.05)
```

```
cor(v1, v2)
```

```
v1unit = v1/sqrt(sum(v1^2))
v2unit = v2/sqrt(sum(v2^2))
```

```
acos(v1unit%*%v2unit) * 180/pi
```

This shows that the two selection gradients are indeed rather different.

In Table 1 there is only p-values, no parameter estimates.

Methods

Line 384: Details missing about the PCA. Based on correlations or covariances? (Line 483 suggests former)

Line 472-473: That is, measured selection?

Line 511: "For these analyzes, fitness traits and PC axes were normalized with a mean of 0 and standard deviation of 1." Does this mean that fitness was z-transformed (scaled to zero mean and unit variance)? Or was fitness relativized (divided by the mean)? Please clarify.

Table S9. No standard errors. Please present complete results throughout, including parameter estimates, standard errors and units.

Dear reviewer,

We appreciate the enthusiasm for our work and are grateful that reviewer 2 could clarify some of their concerns. We feel that adding more details as requested has improved our manuscript considerably. Below we discuss point by point how we have addressed each of the reviewer's concerns.

From the reviewer:

This is my second time reviewing this manuscript. The authors have made a number of changes from the last version, including implementing one of my own suggestions from the last round, computing and reporting selection coefficients for PC axes projected back into the original trait space. I think the manuscript has improved, but I still believe there is a lack of quantification and quantitative interpretation of the results that makes it difficult to judge the meaning of the otherwise interesting results. In the following I have tried to give some very concrete examples of where I think quantification is warranted but currently missing.

We thank the reviewer for bringing these concerns to our attention. We have struggled with finding a balance between long lists of numbers that explicitly give the results in the text and prose that describe the general trends found across the analyses. We have made an effort to add the specific results requested. We have also made every attempt to give all the raw results in tables so effect sizes and significance can be evaluated by the reader.

For example, the section "Adaptive divergence between taxa" in the results completely lack quantification and quantitative interpretation. One example:

"We quantified five fitness-related traits: herbivory, fruit number, flower number, biomass, and survival and find that a significant taxon-by-garden interaction predicts trait values, indicating that the relative success of the species depends on the garden in which they are grown (Fig. 3; Table 1). Adaptive divergence is evidenced by either the local species having higher fitness than the foreign species in the local species' garden, or by a focal species having highest fitness in its home garden compared to all other away gardens."

Here I would want, as a reader, to know how many % higher fitness the local species has compared to foreign species, or by how many % the local species did better in its own habitat.

We have added more explicit descriptions of how much better species do in their home environment compared to other species and in other environments (lines 147-154). We struggled with how much to explicitly tell the numerical results in the text because we have 5 proxies for fitness, three species and three gardens. We felt that reporting all the 60 combinations of effect sizes and significance in the text became cumbersome to read. We have added all the contrast estimates to Table 1 and note that the effect sizes are all graphed in figure 3F. The estimates for fitness of each species in each environment is also reported in figures 3A-E allowing an assessment of the degree of fitness differences we observed.

In the first sentence, I would present biology first and statistics later: "We quantified five fitness-related traits: herbivory, fruit number, flower number, biomass, and survival and find that the relative success of the species depends on the garden in which they are grown, as indicated by statistical support for a taxon-by-garden interaction (Fig. 3; Table 1)"

We have modified the text as suggested by the reviewer.

"amoena had higher survival and experienced significantly less herbivory than both pilosa and deamii, and produced more fruits than pilosa plants. In the deamii habitat, deamii had significantly higher survival than pilosa. In the pilosa

habitat, pilosa plants produced significantly more fruits, and survived at a higher rate than both amoena and deamii plants. “

What is “significantly less”? What is “more fruits”? “A higher rate”? Please tell us what the differences were, not only if they were statistically supported or not.

We have given more quantitative descriptions of the degree of success local plants have in the text and note that the data are all presented in figure 3 and Table 1 so readers can assess the strength and biological importance of each comparison.

Similarly, some examples from the section “Local adaptation within species”:

“Local adaptation was evidenced by a negative relationship between the estimated population effect on fitness and distance of the population from the common garden.”

Please present slope, standard error, and units. “Reproductive performance in terms of fruit set decreased by x% per km distance between the garden site and the natural population (slope = $x \pm y$ fruits/km)”

“Final biomass in pilosa also shows a strong negative correlation with geographic distance”

Here, something like “Biomass decreased by x.xx g per km increase in distance ()”

We have attempted to add more specific numbers of how much fitness changes by distance (line 181-197). Specifically, we added biomass per geographic distance because these have the most logical and intuitive units. It is difficult to do this for genetic distance or environmental distance because these are in either PC units or abstract FST genetic units that do not translate into something intuitive. All slopes are reported in table S4.

Related issues occur in the section “Selection between and within species”, and here there is also substantial ambiguity in how the analyses were done.

“Due to collinearity between traits we summarized phenotypic variation using a principal components analysis (Fig. 5).”

Please state whether the PCA was performed on standardized data (I think it was).

We have clarified in the results section that the PCA was on raw trait data. We also make sure to explain this in the methods section.

“We investigated how leaf trait variation (PC1 and PC2) explained variation in fitness traits (fruit set, flower set, and biomass) in the pilosa garden”

Does this mean a formal phenotypic-selection analysis was done (regressing relative fitness on the traits)? There are contradictory statements here and in the Method. Be explicit.

We have tried to clarify our wording in this paragraph (line 208-212). We note that this is a brief summary of our strategy and the full explanation of the tests we ran and their results are in the subsequent paragraphs. We also walk through the details of these methods in the Methods section of the paper. We are running a series of regression models with relative fitness and PCs of traits.

“For PC2 the interaction term in our model is significant indicating that the relationship between PC2 and fitness traits varied across species (Table S8).”

Selection varied between species, as indicated by support for the interaction term?

We have edited the wording to highlight the biological finding first and then the statistical test results as suggested by the reviewer (line 238)

“The results reveal that fitness differences between species is due to strong selection acting on leaf length, and leaf length/width ratio (leaf shape), whereas within pilosa there is strong selection acting on specific leaf area, leaf area, and chlorophyll content.”

What is “strong selection”? First, we need to know how selection was quantified. I think variance-standardized selection gradients, but not sure (was the response variable relative or absolute fitness? If the values are indeed variance-standardized selection gradients, the interpretation is % change in fitness per SD change in the trait. Quantify and judge strength based on the literature (see e.g. Hereford et al. 2004 Evolution for a thorough discussion about interpretation of the strength of selection).

As above, we struggled with how many lists of numbers to add to the text of the results. All the β values are reported in figure 5h. As suggested by the reviewer we did add the inferred selection coefficients in this paragraph (line 266-271) and removed the qualifications of selection as “strong” as suggested.

To say something about the difference in selection between levels, one method could be to calculate the angle between the selection gradients (if the reported values are formal selection gradients, see above). For the first fitness component (#Flowers), this could be done as follows in R:

```
v1 = c(-0.09, -0.02, 0.12, 0.12, -0.03, 0.16)
```

```
v2 = c(-0.25, 0.16, -0.01, 0.11, 0.07, 0.05)
```

```
cor(v1, v2)
```

```
v1unit = v1/sqrt(sum(v1^2))
```

```
v2unit = v2/sqrt(sum(v2^2))
```

```
acos(v1unit%*%v2unit) * 180/pi
```

This shows that the two selection gradients are indeed rather different.

We appreciate the reviewer suggesting a way to quantify how different the selection gradients are. For all three fitness metrics we find angle between selection coefficients is around 60. After considering this analyses we feel it may be overly complicated and difficult to interpret. We fear it does not have an intuitive biological meaning and is too far removed from the data we collected. We have measured traits, summarized them with a PC analyses, inferred selection on the PCs, back calculated the selection on the traits given the selection on the PC axes and are now trying to determine the degree to which these selection axes are different. We believe that the results of our regression analyses (Figure 5B-G, table S9) and the heat map showing the difference in selection gradients (Figure 5H) is a simpler way of conveying the same information. If this is deemed an important metric to include we are happy to do so but would request a little guidance on how to interpret this angle.

In Table 1 there is only p-values, no parameter estimates.

We have updated this table with parameter estimates.

Line 384: Details missing about the PCA. Based on correlations or covariances? (Line 483 suggests former)

We have added clarification that this is a correlation matrix.

Line 472-473: That is, measured selection?

We have added clarification that this is investigating selection.

Line 511: "For these analyzes, fitness traits and PC axes were normalized with a mean of 0 and standard deviation of 1." Does this mean that fitness was z-transformed (scaled to zero mean and unit variance)? Or was fitness relativized (divided by the mean)? Please clarify.

We have added clarification about how our fitness metrics were z-transformed.

Table S9. No standard errors. Please present complete results throughout, including parameter estimates, standard errors and units

We have added standard errors.

Reviewer #2 (Remarks to the Author):

The authors have made a number changes in response to the last round of comments. There is now some more quantification, which I think helps to clarify the results.

My last "major" quibble is that the current analysis of trait-fitness relationships appears not to be a formal selection analysis, because fitness is z-transformed (scaled to zero mean and unit variance) rather than relativized (divided by the mean). This changes the interpretation of the slopes, which should be made clear. (Or change the scaling and present selection gradients).

Reviewer #2 (Remarks on code availability):

Code appears clean and readable

Dear reviewer,

We have made the requested adjustment to how we transformed our fitness data. We have now relativized the data (divided by the mean of each trait). This has not impacted the qualitative results and all the same patterns and p-values remain the same. We have adjusted all relevant numbers including Table S8 and S9 and figure 5. We have also clarified in the text (both the results and methods) that this change was made.

We believe this final revision has addressed all of the reviewer's concerns.

From reviewer:

The authors have made a number changes in response to the last round of comments. There is now some more quantification, which I think helps to clarify the results.

My last "major" quibble is that the current analysis of trait-fitness relationships appears not to be a formal selection analysis, because fitness is z-transformed (scaled to zero mean and unit variance) rather than relativized (divided by the mean). This changes the interpretation of the slopes, which should be made clear. (Or change the scaling and present selection gradients).